# The Immunometabolic Roles of Various Fatty Acids in Macrophages and Lymphocytes

**DOI:** 10.3390/ijms22168460

**Published:** 2021-08-06

**Authors:** Jose Cesar Rosa Neto, Philip C. Calder, Rui Curi, Philip Newsholme, Jaswinder K. Sethi, Loreana S. Silveira

**Affiliations:** 1Immunometabolism Research Group, Department of Cell Biology and Development, Institute of Biomedical Science, University of Sao Paulo, Sao Paulo 05508-000, Brazil; loreana_loly@hotmail.com; 2LIM-26, Hospital das Clínicas of the University of São Paulo, Sao Paulo 01246-903, Brazil; 3Faculty of Medicine, School of Human Development and Health, University of Southampton, Southampton SO16 6YD, UK; p.c.calder@soton.ac.uk (P.C.C.); j.sethi@soton.ac.uk (J.K.S.); 4National Institute for Health Research Southampton Biomedical Research Centre, University of Southampton and University Hospital Southampton National Health Service (NHS) Foundation Trust, Southampton SO16 6YD, UK; 5Institute for Life Sciences, University of Southampton, Southampton SO17 1BJ, UK; 6Interdisciplinary Post-Graduate Program in Health Sciences, Cruzeiro do Sul University, Sao Paulo 01506-000, Brazil; ruicuri59@gmail.com; 7Curtin Medical School and Curtin Health Innovation Research Institute, Curtin University, Perth, WA 6102, Australia; philip.newsholme@curtin.edu.au

**Keywords:** immune cells, lipids, fatty acids, metabolism, leukocytes, macrophages, lymphocytes, inflammation, cytokines

## Abstract

Macrophages and lymphocytes demonstrate metabolic plasticity, which is dependent partly on their state of activation and partly on the availability of various energy yielding and biosynthetic substrates (fatty acids, glucose, and amino acids). These substrates are essential to fuel-based metabolic reprogramming that supports optimal immune function, including the inflammatory response. In this review, we will focus on metabolism in macrophages and lymphocytes and discuss the role of fatty acids in governing the phenotype, activation, and functional status of these important cells. We summarize the current understanding of the pathways of fatty acid metabolism and related mechanisms of action and also explore possible new perspectives in this exciting area of research.

## 1. Introduction

Dietary fatty acids (FAs) are widely known to affect metabolic and cardiovascular health with different fatty acids having diverse effects [1]. In general, polyunsaturated fatty acids (PUFAs) are known to reduce the risk of cardiovascular disease [2] and to have therapeutic effects in both cardiovascular [3] and inflammatory diseases [4]. Beyond their effects on physiology, it is important to explore the molecular and cellular mechanisms of action of specific dietary FAs in order to better understand how they influence human health and wellbeing. The functioning of the immune system, for example, is well known to be intimately linked to FA and nutrient availability and energy status more generally [5,6]. The basis of this link can be found at the cellular level, where the combination of nutrient availability and metabolic status supports and often determines the spectrum of energy-demanding activities that can be performed by distinct immune cell types, so enabling their functional capacity. Moreover, metabolic plasticity may be key to determining the ability of the immune system to adapt to a constantly changing microenvironment [7].

FAs have a hydrocarbon chain with a carboxyl group at one end and a methyl group at the other end of the chain [8]. The chain may vary in length and in degree of unsaturation (i.e., presence and number of double bonds). The naming of FAs relates to their structural features: they are described as short, medium, long, or very long depending upon chain length and saturated, monounsaturated, or polyunsaturated depending upon their degree of unsaturation [8]. There are two main classes of PUFAs called omega-3 and omega-6, also referred to as n-3 or n-6, respectively. The omega carbon is identified as the farthest from the carboxyl group [8]. The names, structures, and symbols of the fatty acids mentioned in this review are listed in Table 1.

Different subsets of immune cells appear to have differential metabolic profiles [9,10,11]. In other words, the metabolic signature of immune cells may vary according to their subtype, likely due to the necessity to generate adequate levels of energy in order to survive and for the production of intermediates in an attempt to maintain growth, proliferation, and function. Thus, there are likely to be alterations in the operation of different metabolic pathways—glycolysis, the citric acid (tricarboxylic acid (TCA) or Krebs) cycle, the pentose phosphate pathway (PPP), FA oxidation, FA synthesis, amino acid metabolism, and oxidative phosphorylation—to support the functional capacity of different immune cell subsets [12]. Macrophages are innate immune cells. As part of the first line of immune defense, the role of macrophages is to respond rapidly and appropriately to a variety of agents or physiological challenges. For this, they need to maintain a flexible functionality [13,14]. In general, they can be activated into pro-inflammatory macrophages (classically activated; formerly called M1) or anti-inflammatory macrophages (alternatively activated; formerly called M2) but there is a broad spectrum between these subtypes along with a broad range of activating stimuli [13,14,15]. The metabolic profiles of these macrophage subtypes are distinct. Pro-inflammatory macrophages generate ATP primarily from enhanced glycolysis, and biomaterials from enhanced PPP, glutaminolysis, inducible nitric oxide synthase-mediated arginine metabolism and lipogenesis [7]. Collectively this metabolic program supports the main microbicidal and tumoricidal roles of pro-inflammatory macrophages. Such macrophages can cause host tissue damage. In contrast, anti-inflammatory macrophages generate ATP primarily by enhanced oxidative phosphorylation and by β-oxidation of FAs, and biomaterials from proline and polyamine synthesis via arginase-dependent metabolism [16]. This program supports the roles of anti-inflammatory macrophages in wound repair, tissue remodeling, and tumor growth. There is evidence that FA oxidation induces the anti-inflammatory macrophage phenotype [11]. Although the necessity for FA oxidation to enable induction of anti-inflammatory polarization is still unclear [17], deletion of FA transporter proteins induces the pro-inflammatory phenotype in macrophages [18], indicating that FA utilization can reduce pro-inflammatory polarization.

T and B lymphocytes are part of adaptive immunity, while innate lymphoid cells and natural killer cells are part of the innate immune system. B cells produce antibodies while T cells have multiple roles in anti-bacterial, anti-viral, and anti-tumor defense according to specialized subsets [19]. There are several T cell subsets defined according to cell surface markers and cytokine profiles and functions [19]. In general, T cells are classified as CD4^+^ (T helper; Th) or CD8^+^. CD4^+^ T cells may be further classified as naïve, Th1, Th2, T regulatory (Treg), Th17, or memory, while CD8^+^ T cells are further classified as naïve, cytotoxic, memory, or effector memory [19]. B lymphocytes are classified as naïve, memory, or plasma cells [19].

Immature T cells are found in the thymus and are initially CD4 and CD8 double negative. During their maturation they become both CD4 and CD8 double positive and then lose one or the other to produce naïve CD4^+^ or CD8^+^ T cells. Clonal expansion of T cells following rearrangement of the T cell receptor (TCR), which occurs as part of T cell maturation, requires energy and this is generated by glycolysis with glucose influx into the cells involving glucose transporter (GLUT)-1 [10]. Conversely, naïve T cells themselves revert to a low-energy state with a significant reduction in glucose transport [20].

Under resting conditions, T and B lymphocyte metabolism are distinct. Resting T lymphocytes have higher glucose uptake and ATP generation than resting B cells [21], which have higher mitochondrial mass and are more dependent on oxidative metabolism [22].

Regardless of the type of immune cell, both extracellular and intracellular FAs strongly influence their functionality. Extracellular FAs can be recognized and/or taken up by a range of cell membrane proteins including G protein-coupled receptors, scavenger receptor CD36, fatty acid-binding proteins, or members of the fatty acid transport protein (FATP) family. Once inside the cell, FAs can be fuels for β-oxidation, substrates for complex lipid synthesis (e.g., phospholipids for cell membranes), substrates for lipid mediator synthesis, or signals for nuclear receptors that control expression of genes related to metabolism and inflammation amongst other processes [23,24].

Upon activation of Th1 lymphocytes, aerobic glycolysis predominates to sustain the rapid increase in demand for energy and biosynthetic intermediates [25]. In contrast, Th2 cells show high rates of FA oxidation [26]. Regulatory and memory T cells, which are not proliferative, also show a dependence on FA oxidation [7]. Finally, Th17 lymphocytes have a high capacity to produce ATP from glycolysis [27].

Thus, FA oxidation is important in some immune cell subsets and there seems to be a link between switching to FA oxidation and immune cell function [24]. We will now focus on FA metabolism in lymphocytes and macrophages and address the role of different FAs in the metabolism of these cells and the consequences of lipid overload. Finally, we will propose new horizons to clarify the role of FAs in immunometabolism.

## 2. Fatty Acids and Macrophages

### 2.1. Fatty Acid Metabolism in Macrophage Subsets

For an overview on macrophage metabolism, see Curi et al. [28]. As mentioned above, the most studied macrophage subsets are the classically activated (M1) phenotype and the alternatively activated (M2) phenotype. M1 macrophages are related to the Th1 lymphocyte response, type I inflammation, killing of intracellular pathogens and tumor resistance, while M2 macrophages are related to Th2 lymphocyte responses/activation, allergy, immune regulation, tissue remodeling, killing parasites, and tumor promotion [15]. M1 macrophages exhibit high glycolytic activity and the overexpression of GLUT-1 can induce classical activation of monocytes to the M1 subtype [29]. During M1 activation, the cytosolic accumulation of citrate allows for the production of specific lipid mediators by de novo lipogenesis [30]. By decreasing FA oxidation, M1 macrophages accumulate triacylglycerol and cholesterol-rich lipid droplets [31,32]. Thus, M1 macrophages have lower FA uptake and oxidation than M2 macrophages but raised de novo lipogenesis [33]. Activating pro-inflammatory cytokines have been reported to decrease the level of FATP, thereby decreasing FA uptake [18]; this provides a link between pro-inflammatory stimuli and macrophage metabolism. Figure 1 describes some key metabolic differences between M1 and M2 macrophages.

FA oxidation is necessary to support the functions of alternatively activated macrophages [34]. FATP-1 deletion, specifically in myeloid cells, reduces FA oxidation and increases glycolytic flux in macrophages [18]. Mice fed a high-fat diet exhibit impaired whole-body glucose metabolism and higher inflammation in adipose tissue with the deletion of FATP-1 in myeloid cells [18]. Overexpression of FATP-1 in RAW267 macrophages reduces M1 polarization by modulating lipid mediators: macrophages with FATP-1 overexpression had lower arachidonic acid synthesis and increased levels of arachidonic acid metabolites 15-hydroxyeicosatetraenoic acid (HETE) and 12-HETE [18].

In mammalian cells, one important lipogenic transcription factor is peroxisome proliferator activated receptor (PPAR)γ [35]. Several lipids such as docosahexaenoic acid (DHA), eicosapentaenoic acid (EPA), and palmitic acid can activate this transcription factor [36,37,38]. Xu et al. reported that 12- and 15-HETE induce PPARγ activation [35]. PPARγ promotes anti-inflammatory actions and is crucial for the differentiation to alternatively activated macrophages [39]. Consistent with this, the deletion of PPARγ in myeloid cells has been shown to reduce the expression of CD206, a surface marker of M2 macrophages [40]. However, in the same study of obese mice, the anti-inflammatory response caused by aerobic training was maintained in adipose tissue by restoring adipocyte area and reducing tumor necrosis factor (TNF)-α and interleukin (IL)-1β secretion [40]. Moreover, mice with the same deletion of PPARγ in myeloid cells demonstrated impaired muscle regeneration as a result of a defective remodeling phase after injury [41]. Thus, PPARγ expression in myeloid cells is required to activate the phenotype of macrophages necessary for skeletal muscle repair after injury [41].

### 2.2. Fatty Acid Overload in Macrophages

The availability of specific types of FAs can lead to altered cell membrane composition and biophysical properties. The treatment of human macrophages with saturated FAs modified the phospholipid FA composition and induced an increase in the saturated/unsaturated ratio in cell phospholipids [42]. The high content of saturated FAs, especially stearic acid, in phospholipids reduced the membrane fluidity and promoted the impairment of Na^+^/K^+^-ATPase, leading to the elevation of intracellular K^+^ levels [42]. This effect was counteracted by co-treatment with the monounsaturated FA oleic acid [42]. Poledne et al. [43] related adipocyte phospholipid FAs to macrophage subtypes present in human adipose tissue. They found that palmitic and palmitoleic acids in adipocyte phospholipids were positively associated with M1 macrophages while stearic acid, α-linolenic acid, EPA, total n-3 FAs, and the ratio of n-3 to n-6 FAs were inversely associated. Conversely, stearic acid, α-linolenic acid, EPA, total n-3 FAs, and the ratio of n-3 to n-6 FAs were positively associated with M2 macrophages and palmitoleic acid was inversely associated. The positive relationship between palmitic acid and M1 macrophages fits with earlier data showing that saturated FAs, including palmitic acid, enhance inflammation [44,45,46,47]. Palmitic acid may stimulate macrophages directly via Toll-like receptor (TLR)-4 [37,44]. However, palmitic acid also reduces mitochondrial adenine nucleotide translocator activity, increasing intracellular ATP content and reducing adenosine 5′ monophosphate-activated protein kinase (AMPK) activity [37]. The positive relationship of omega-3 FAs with M2 macrophages fits with the generally accepted anti-inflammatory effects of these fatty acids [48,49]. However, their relationships with palmitoleic acid are unclear, since this FA exhibits an anti-inflammatory effect in vitro [50,51], which seems to involve AMPK activation [50]. On the other hand, palmitoleic acid promotes the differentiation of macrophages into the M2 phenotype independent of the lipid sensor PPARγ [52]. In this sense, the balance between FA oxidation and FA synthesis may be essential to macrophage polarization and function. There is increased FA production by acyl-CoA synthetase-1 (ACS-1) in macrophages stimulated with lipopolysaccharide (LPS) and knockdown of ACS-1 reduces the inflammatory response after LPS treatment [53].

Oxidized low-density lipoproteins that are taken up by CD36 promote mitochondrial reprogramming in macrophages characterized by lower FA oxidation and enhanced aerobic glycolysis [54]. The mechanism by which the oxidized low-density lipoproteins induce this reprogramming is still unclear, but macrophage exposure to oxidized phospholipids in oxidized low-density lipoproteins increases aerobic glycolysis [54].

In humans, macrophages derived from monocytes showed an association between membrane lipid composition and polarization state with specific phospholipid remodeling within the M2a and M2c subsets [55]. A question that arises in the context of FAs orchestrating immune cell phenotypes is whether circulating cells in people with obesity are affected by the enhanced exposure to lipids in plasma. Ghanim et al. studied mononuclear cells from healthy non-diabetic individuals with obesity. They concluded that these cells contribute to elevated pro-inflammatory cytokine concentrations, including TNF-α and IL-6, via enhanced nuclear factor kappa-light-chain-enhancer of activated B cells (NFκB) transcriptional activity [56]. One aspect of the anti-inflammatory actions of omega-3 FAs could be attenuation of macrophage differentiation towards more inflammatory phenotypes. The treatment of obese people with α-linolenic acid (4 g/daily for 12 weeks) decreased plasma free FAs and pro-inflammatory cytokines and increased PPARγ mRNA expression in mononuclear cells [57]. Adolescents with obesity who received 1.2 g/day of omega-3 FAs for the same period of intervention had elevated serum omega-3 FA concentrations together with lower inflammatory markers [58]. Apart from the control of macrophage metabolism and function by FAs, Lau et al. [59] reported that monocyte-derived macrophages from patients with Alzheimer’s disease treated with omega-3 FAs had increased ATP synthesis through oxidative phosphorylation [59], suggesting a move to a less inflammatory phenotype.

### 2.3. Fatty Acid-Stimulated Regulation of Macrophage Function

The link between high fat diet-induced obesity and chronic low-grade inflammation involves an increased pro-inflammatory state, including M1 macrophage polarization [60]. We review here the effects of different FAs on three principal macrophage functions: chemokine production, phagocytosis and adhesion.

Treatment of adipocytes with saturated FAs (lauric, myristic, and palmitic acids) increased the expression of the chemokine CCL2 (monocyte chemoattractant protein 1 (MCP-1)) [61], which induces the chemoattraction of non-resident monocytes. Co-treatment with the omega-3 FA DHA reversed the effect of saturated FAs in a PPARγ-dependent manner [61]. Similarly, treatment of macrophages with palmitic acid stimulated MCP-1 and IL-8 expression [62]. Palmitic acid treatment increased macrophage production of IL-8 and CXCL1 [63]. Conversely, macrophages treated with palmitoleic acid exhibited reduced LPS-induced MCP-1 expression [51]. Co-treatment with oleic acid and LPS did not blunt MCP-1 protein expression in macrophages [64]. The omega-3 FAs EPA and DHA (100 μM) reduced MCP-1 production by LPS-stimulated monocytes [65] and MCP-1 and IL-8 production by TNF-stimulated endothelial cells (25 μM or 50 μM) [66]. DHA also decreased regulated, normal T cell expression and presumably secreted (RANTES) production by TNF-stimulated endothelial cells upon activation [66]. Specific FAs have been shown to modulate macrophage phagocytosis. For example, PUFAs enhanced phagocytosis [67,68], while saturated FAs impair phagocytosis, an effect related to the fatty acid composition of the cell membrane [69]. The effects of FAs, particularly omega-3 FAs, on adhesion molecules and leukocyte-endothelial adhesion were reviewed recently [70]. Both EPA and DHA can decrease the expression of key adhesion molecules, such as vascular cell adhesion molecule 1, and result in reduced adhesive interactions between leukocytes and endothelial cells.

### 2.4. Fatty Acids as Signaling Molecules in Macrophages

Saturated FAs are co-regulators of LPS/TLR-4 activation, and co-treatment with palmitic acid and LPS induces a substantial inflammatory response [71]. However, the reason for saturated FAs co-stimulating signaling via LPS/TLR-4 is unclear. Rowe et al. reported that the myristoylation of TIR-domain-containing adapter-inducing interferon-β (TRIF)-related adaptor molecule is an essential step in signal transduction within the LPS/TLR-4 pathway [72]. Furthermore, the conjugation of palmitic acid with cysteine 113 and 274 residues of myeloid differentiation primary response 88 (MYD88) is necessary to activate the LPS/TLR-4 downstream activities [73]. The activation of CD36 and fatty acid synthase are essential to this palmitoylation of MYD88 [73]. This may be why palmitic acid co-stimulates the LPS/TLR-4 system. The anti-inflammatory effects of palmitoleic acid may be due to a reduction in CD36 expression and subsequent decline in FA uptake [52]. Molecular docking experiments showed that palmitoleic acid has a high affinity with the F126 loop site of MD2, a TLR-4 adaptor essential for canonical activation of the TLR-4 pathway. The binding between palmitoleic acid and the F126 loop modifies the structure of MD-2 protein and reduces the docking between TLR-4/MD2 [52].

Another relevant mechanism is the role of FAs in membrane lipid raft formation underpinning pro-inflammatory and anti-inflammatory pathways. TLR-4 stabilization by lipid rafts rich in cholesterol is considered important in the activation of LPS/TLR-4 [74]. Saturated FAs were demonstrated to promote raft formation in LPS-stimulated cells [44,46], an effect that was prevented by the omega-3 FA DHA. LPS/TLR-4 affects cell membrane lipid composition and fluidity, probably by a fatty acid synthase-dependent mechanism [74,75]. The concept of modification in membrane fluidity and macrophage function has been under debate for a long time. Calder et al. discussed the role of PUFAs in enhanced macrophage phagocytosis associated with this concept [69]. Omega-3 and omega-6 PUFAs appear to have differential incorporation into membrane lipids and therefore differentially affect lipid raft formation, altering inflammatory responses [76]. However, there are still gaps in our understanding of the mechanisms of incorporation of FAs and their effects on lipid rafts and inflammatory signaling in macrophages.

Different lipid classes have a crucial role in the inflammatory response and the role of FA-mediated signaling induced by different lipid mediators, like prostaglandins, leukotrienes, maresins, and resolvins, is gaining increased attention. Metabolomics and lipidomics now allow the investigation of the involvement of many other lipid classes in macrophage inflammatory responses [77]. It is known that sphingolipids, ceramides, phospholipids, lyso-phospholipids, and endocannabinoids can modify macrophage responses. For example, the formation of ceramides from neutral sphingomyelinase-2 is a crucial step in the TNF-α response of macrophages [78]. The role of different FAs in the formation, degradation, and activity of these different endogenous lipids in the context of macrophage function is an exciting avenue to investigate.

Thus, classical and alternative macrophages show differences in FA metabolism. This difference is important for supplying the correct metabolic intermediaries to different stages of activation of macrophages, supporting their functions. Moreover, the different lipid classes and the individual FAs themselves can modulate the immune response. Therefore, both FA metabolism and the type of FAs available to cells can be relevant to induction or inhibition of the immuno-inflammatory response.

## 3. Fatty Acids and Lymphocytes

### 3.1. Summary of Fatty Acid Metabolism in Lymphocytes

Lymphocytes utilize glucose and glutamine at high rates [79]. Burns et al. [80] reported that palmitic acid oxidation was higher in lymphocytes than neutrophils. The membrane FA composition and fluidity and the activation of membrane proteins vary according to the activation state of lymphocytes [81,82].

### 3.2. CD4 T Lymphocytes and Fatty Acid Metabolism

Different subsets of CD4^+^ T cells have different metabolic signatures. T cell activation occurs by TCR recognition of ligands presented by major histocompatibility complex after exposure to an antigen-presenting cell and is associated with co-stimulatory signaling by CD28. As previously mentioned, this step of T cell activation requires aerobic glycolysis for rapid ATP production together with the Warburg effect necessary for nucleotide generation, lipid synthesis, and biomass induction [83]. The primary regulators for these mechanisms are the mechanistic target of rapamycin (mTOR) induced downstream via phosphoinositide 3-kinase-protein kinase B (PI3K-AKT) signaling and c-Myc [84].

After the activation of T lymphocytes, the flux of carbon generated by the TCA cycle is redirected to lipid synthesis: excess citrate produced in the TCA cycle is transported from the mitochondria to the cytosol, where it is converted to acetyl-CoA, elongated to malonyl-CoA by acetyl CoA carboxylase (ACC), and consequently used for FA, cholesterol, and triglyceride biosynthesis [83]. ACC is an essential enzyme for T cell expansion [85]. In vitro studies have shown that FA uptake increases during T cell proliferation [86]. During the activation of T effector cells, mitochondrial fission reduces electron transport chain efficiency, a mechanism regulated by dynamin related protein-1 and necessary to induce the Warburg effect, decreasing the rate of FA oxidation [87].

During CD4 cell activation, there is an increase in the content of palmitic acid in three lipids classes: diacylglycerol, phosphatidylethanolamine, and hexosylceramides [88]. Arachidonic, palmitic, palmitoleic, and myristic acids were all higher in triacylglycerols after stimulation of CD4 T cells [88]. 

Regarding CD4 T cell subsets, Th1, Th2, and Th17 lymphocytes have the greatest ATP production from glycolysis, with a considerable increase in GLUT-1 expression sustained by mTOR complex (mTORC) 1 activation [25]. In contrast, ATP production in Treg cells mainly occurs through FA oxidation [24,25] where the high AMPK activity regulates FA oxidation [24]. Furthermore, in Tregs the mTORC pathway is inhibited by phosphatase and tensin homolog (PTEN) [89]; PTEN is a phosphatase that is able to dephosphorylate phosphoinositol 3-phosphate, which is essential to activation of the AKT/mTORC pathway [89]. Memory T lymphocytes primarily use FA oxidation for ATP production [7].

The differentiation from CD4 T naïve cells to CD4 Th17 cells and the activation of Th17 cells are both dependent on FA synthesis [90]. Berod et al. showed that the inhibition of ACC reduces Th17 differentiation and induces Treg differentiation [91]. Glycolyis is a key step involved in the induction of Th17 cells, an increase in long chain acylcarnitine, and a reduction in β-oxidation occuring together with Th17 differentiation and activation [92].

Hradilkova et al. reported that re-stimulated Th1 lymphocytes presented increased FA oxidation and reduced glycolysis [93]. This metabolic feature associates with Twist-1 expression that promotes FA oxidation and protects the lymphocytes against oxidative stress [93].

Interestingly, follicular helper T cells, which are essential for the activation of B lymphocytes and humoral immunity, require the activation of phosphoethanolamine synthesis as a key step for differentiation [94]. 

### 3.3. CD8 T Lymphocytes and Fatty Acid Metabolism

Activation of CD8^+^ T cells induces metabolic reprogramming mainly towards increased aerobic glycolysis [25] for which CD28 co-stimulation is necessary [95]. Thus, the intermediates of the TCA cycle are used in biosynthetic pathways. Regarding FA metabolism, de novo lipogenesis is essential to sustaining CD8 effector T cell activity [96]. In general, FA oxidation is unnecessary to this subset [97]. Nonetheless, programmed cell death protein 1 (PD-1) signaling can switch the metabolic signature of these cells to FA oxidation [98]. Moreover, CD8^+^ effector cells show mitochondrial fission, and inhibition of proteins that regulate this fission leads to the differentiation of CD8 cells to memory cells [87].

CD8 memory T cells have reduced mTORC1 activity and use FA oxidation as the primary ATP source [97]. Thus, mitochondrial biogenesis is necessary to meet the energy demands of CD8 memory cells [87,99]. Nevertheless, Sukumar et al. described that the induction of glycolysis during the differentiation of CD8 memory cells leads to short-term CD8 memory [100]. On the other hand, the inhibition of glycolysis during memory T cell development induces a robust long-term CD8 memory phenotype [100]. The metabolic reprogramming of the CD8 memory subset is induced by IL-15, which induces the mitochondrial transcription factor A to promote mitochondrial biogenesis [101]. The primary site of lipolysis in the CD8 memory subset is lysosomes [102]. Moreover, CD8 memory cells, mainly the central memory T lymphocytes found in lymph nodes, can store triglycerides, while the source of FAs for oxidation in skin CD8 memory T cells is extracellular [97,103]. Finally, a higher mitochondrial mass is a metabolic advantage when CD8 memory cells are re-stimulated because of the increased ATP demand which is met by a rapid increase in glycolysis and higher oxidative metabolism [104].

### 3.4. B Lymphocytes and Fatty Acid Metabolism

B lymphocytes play a role in the humoral response, producing antibodies and a memory response. After B cell receptor (BCR) engagement, together with co-stimulatory signals, B cells undergo rapid clonal expansion. Like T cells, this step depends on glycolysis and regulation by the PI3K-AKT-mTORC1 axis [9]. Glutamine uptake and metabolism increase after BCR activation, as does lipid biosynthesis. However, unlike in T cells, there is higher coordination between oxidation of glucose and FAs and glycolysis [9].

In plasma cells, which are crucial for antibody production, glycolysis and oxidative glucose metabolism are activated in order to support increased mitochondrial capacity associated with the biosynthesis of antibodies [105,106]. In addition to their high mitochondrial mass, plasma cells exhibit increased autophagy: AMPK, a regulator of autophagy, is essential to generate long-term memory B cells [107,108].

The microanatomical structures found in follicles of germinal centers control B lymphocyte metabolism and long-term immunity [109]. In germinal centers, B cells are formed and memory B cells and long-lived plasma cells are maintained [110]. As observed for T lymphocytes, the glycolytic response is predominant in the proliferation phase of B cells in germinal centers [111]. However, Weisel et al. recently reported that FA oxidation in mitochondria and peroxisomes is the key metabolic pathway during the development and survival of B cells in germinal centers [112].

In summary, resting B cells have a lower production rate of ATP than T lymphocytes [21]. After activation, there is an increase in glycolytic flux, and the requirement for glucose and FA oxidation is higher than for T cells. During activation and antibody production, plasma cells increase flux through glycolysis, but long-term memory B lymphocytes are dependent on mitochondrial mass, autophagy, and FA oxidation. The metabolism of B cells in germinal centers is still unclear, but recently a high need for FA oxidation has been documented [112]. The current understanding of B lymphocyte metabolism is less complete than the knowledge of T lymphocyte metabolism. Some key metabolic pathways in activated B and T cells are described in Figure 2.

### 3.5. Fatty Acid-Stimulated Regulation and Overload in Lymphocytes

The type of FA in the extracellular environment and taken up can result in different functional effects in lymphocytes. EPA and DHA can reduce CD4^+^ T cell activation and chemokinesis in allograft immunization mice [113]. Besides this, in vitro treatment of CD4^+^ T lymphocytes with omega-3 FAs modified the membrane microdomains to reduce activation [113]. CD4^+^ T cells treated with EPA and DHA demonstrated a reduction in activity after antigen presentation [114].

The treatment of antigen presenting cells with palmitic acid impairs antigen presentation to CD8 lymphocytes [115], an effect which is mitigated by co-treatment with oleic acid [115]. Lymphocytes treated with palmitoleic acid had reduced proliferation, CD28 surface expression, and pro-inflammatory cytokine production (interferon-γ, IL-17, IL-6 and TNF-α), whereas IL-4, IL-10, and IL-2 production increased [116]. The palmitoleic acid/palmitic acid ratio positively correlated with PD-1 expression on the surface of CD4^+^ cells [116]. Non-alcoholic fatty liver disease progression is associated with a low palmitoleic acid/palmitic acid ratio and low expression of PD-1 on CD4^+^ lymphocytes [117]. Interestingly PD-1 activation in T lymphocytes induces metabolic reprogramming with enhancement of FA oxidation and is associated with exhausted lymphocytes [98].

The interaction between microbiota-derived short-chain fatty acids and lymphocytes has been investigated mainly in the context of inflammatory bowel diseases. The production of acetate is essential to the effectiveness of memory CD8 cells [118]. Of note, acetate can sustain acetyl-CoA formation in glucose-restricted conditions and maintain the effector function of CD8^+^ cells [119]. Short-chain fatty acids can induce IL-22 production by CD4^+^ T cells [120]; IL-22 is essential to mucosal immunity and the gut epithelial barrier.

In mice, lipid-rich diets elevated chemokine expression by lymphatic endothelial cells promoting T lymphocyte movement to lymph nodes [121]. This effect may be mediated by palmitic acid as suggested in a study with a co-culture system of lymphocytes and lymphatic endothelial cells [121].

People with so-called healthy morbid obesity have alterations in peripheral T cell subsets with higher CD4^+^ cells than their normal-weight comparators [122]. This enhanced CD4^+^ population in people with obesity is composed mainly of T naïve, T central memory, and Th2 cells [123]. However, the mechanisms that may explain this contrast found in T cells from people with obesity (higher CD8^+^ cells in adipose tissue and higher CD4^+^ cells in peripheral blood) are still unclear. In a pilot study, the B cell response in people with obesity was affected by 12 weeks of supplementation with fish oil, a source of EPA and DHA, resulting in lower percentages of memory and plasma cells, supporting the idea that these FAs interfere with T cell activation [124]. Thus, the type of FA and lipid overload both regulate the function of lymphocytes [125]. However, more studies are necessary to determine further details of the effects of different FAs on T and B lymphocyte function.

In summary, the metabolism of FAs is essential to T and B cell polarization and function. Increased de novo lipogenesis is associated with a pro-inflammatory phenotype and with Th1, Th17, and plasma cells. On the other hand, FA oxidation induces tolerogenic and memory responses with increased Treg and T and B memory cells. Moreover, the type of FA present is able to influence the profile of lipid mediators and can modify the immune response. Monounsaturated fatty acids and n-3 PUFAs promote regulatory and suppressive functions in T and B lymphocytes while saturated and n-6 PUFAs promote the inflammatory pathway. Thus, the control of FA metabolism is essential to fine-tune lymphocyte function.

## 4. Concluding Remarks

In summary, a complex link between FA metabolism and the role of different FAs in regulating lymphocyte and macrophage functions has been revealed but requires further investigation. Briefly, FA oxidation is essential to the alternative/regulatory phenotype of macrophages and lymphocytes. On the other hand, de novo lipogenesis and FA synthesis are associated with rapid clonal expansion and a pro-inflammatory phenotype in macrophages and lymphocytes. Thus, the different pathways of FA metabolism are keys to immune cell differentiation and function, but key regulatory steps remain to be identified. With further clarification, new understandings of how cell-specific immunometabolic programs support exact biological responses against pathogens and tumors, and in immune-aging and autoimmune diseases, could be revealed. This could facilitate the development of new therapeutic approaches, by action directly on metabolic pathways or by immunonutrition with intake of FAs to promote a more optimal immune response (e.g., increase in omega-3 FAs to reduce chronic inflammation). The delivery of specific FAs to specific immune cells may be a limitation in developing such targeted therapies. However, using gene editing it may be possible that the silencing or hyperexpression of metabolic pathways of immune cells can be used to improve immune responses.

## Figures and Tables

**Figure 1 ijms-22-08460-f001:**
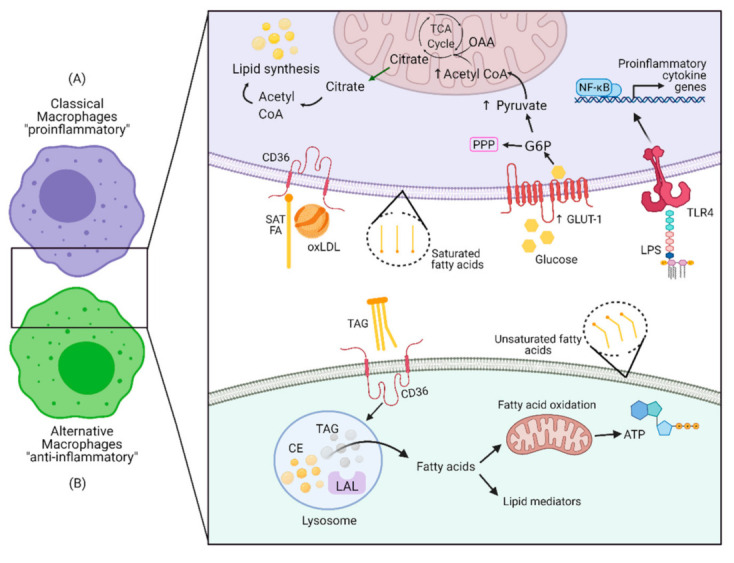
Overview of macrophage metabolism. (**A**) Classically activated macrophages (proinflammatory macrophages [M1]) present a more glycolytic metabolism, even in the presence of oxygen. Their increased GLUT-1 transporter enables higher glucose influx and enhances pentose phosphate pathway (PPP) activity, which is essential for the inflammatory phenotype. The incomplete TCA cycle generates excess citrate, which is crucial for de novo lipid synthesis. The membrane of M1 macrophages has a high content of saturated fatty acids, especially palmitic acid. LPS binding to TLR-4 activates nuclear factor kappa B (NF-κB), responsible for transcription of proinflammatory cytokine genes. (**B**) Metabolism in alternatively activated macrophages (anti-inflammatory macrophages [M2]) mainly produces ATP via the TCA cycle and oxidative phosphorylation (OXPHOS) fueled by fatty acid β-oxidation. There is low glycolysis and PPP activity in this subtype of macrophage. Abbreviations: CD: cluster of differentiation; LAL: lysosomal acid lipase; OAA: oxaloacetate; TCA: tricarboxylic acid cycle; G6P: glucose-6-phosphate; TLR-4: Toll-like receptor 4; SAT: saturated fatty acids; FA: fatty acid; oxLDL: oxidized low-density lipoprotein; GLUT-1, glucose transporter 1; TAG: triacylglycerol; LPS: lipopolysaccharide; CE: cholesteryl ester; ATP: adenosine triphosphate. Created with biorender.com.

**Figure 2 ijms-22-08460-f002:**
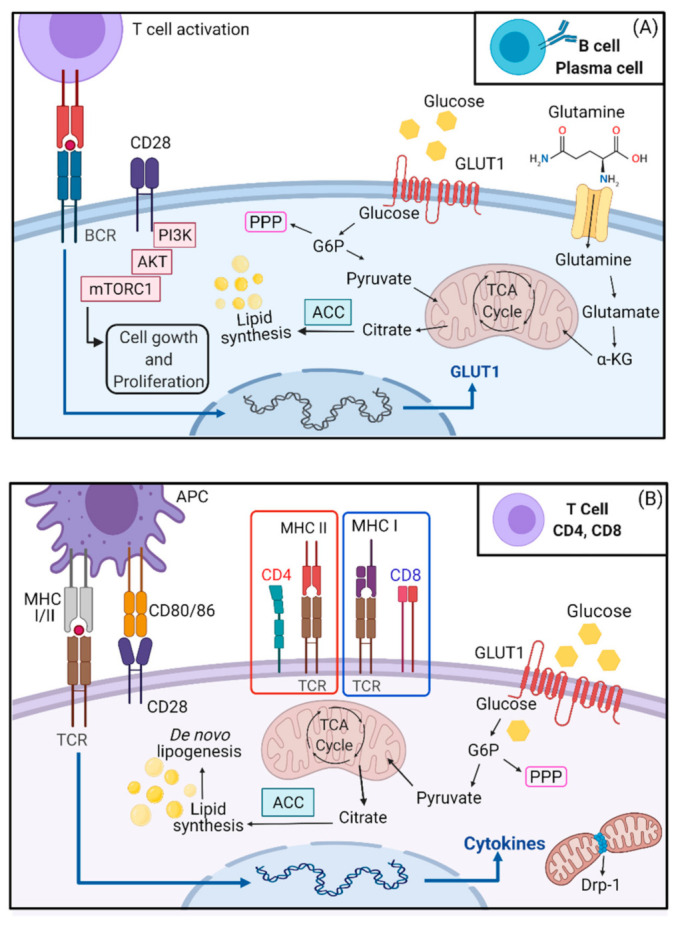
B and T cell activation and metabolism. (**A**) B cells (plasma cells) are activated by a T cell via BCR. Transcription of GLUT-1 is increased and favors glucose uptake and aerobic glycolysis. The pentose phosphate pathway (PPP) is required for redox balance and nucleotide synthesis whereas pyruvate and αKG go to the TCA cycle. Citrate excess is redirected to lipid synthesis. CD28 activation is required for long-lived plasma cells by stimulating growth and proliferation. (**B**) T cell activation occurs via APCs, in which MHC I or II binds to TCR and the co-stimulatory CD28. Drp-1 regulates mitochondrial fission playing an important role in metabolic reprogramming, proliferation, and migration in activated T cells. Glycolysis is upregulated and most of the pyruvate is excreted from the cell as lactate. Citrate from the TCA cycle is used for lipid synthesis. The PPP is important for generating NADPH and nucleotide synthesis. Abbreviations: CD: cluster of differentiation; TCA: tricarboxylic acid cycle; G6P: glucose-6-phosphate; GLUT-1: glucose transporter; Drp-1: dynamin related protein-1; TCR: T cell receptor; MHC: major histocompatibility complex; APC: antigen presenting cell; ACC: acetyl-CoA carboxylase. Created with biorender.com.

**Table 1 ijms-22-08460-t001:** Names, chemical structures, and symbols of the fatty acids mentioned in this review.

Common Name	IUPAC Name	Chemical Structure	Symbol
Lauric acid	Dodecanoic acid	CH_3_(CH_2_)_10_COOH	12:0
Myristic acid	Tetradecanoic acid	CH_3_(CH_2_)_12_COOH	14:0
Palmitic acid	Hexadecanoic acid	CH_3_(CH_2_)_14_COOH	16:0
Stearic acid	Octadecanoic acid	CH_3_(CH_2_)_16_COOH	18:0
Palmitoleic acid	(Z)-hexadec-9-enoic acid	CH_3_(CH_2_)_5_CH=CH(CH_2_)_7_COOH	16:1*n*−7
Oleic acid	(Z)-octadec-9-enoic acid	CH_3_(CH_2_)_7_CH=CH(CH_2_)_7_COOH	18:1*n*−9
α-Linolenic acid	(9Z,12Z,15Z)-octadeca- 9,12,15-trienoic acid	CH_3_CH_2_CH=CHCH_2_CH=CHCH_2_CH=CH(CH_2_)_7_COOH	18:3*n*−3
Eicosapentaenoic acid	(5Z,8Z,11Z,14Z,17Z)-icosa- 5,8,11,14,17-pentaenoic acid	CH_3_CH_2_CH=CHCH_2_CH=CHCH_2_ CH=CHCH_2_CH=CHCH_2_CH=CH(CH_2_)_3_COOH	20:5*n*−3
Docosahexaenoic acid	(4Z,7Z,10Z,13Z,16Z,19Z)-docosa- 4,7,10,13,16,19-hexaenoic acid	CH_3_CH_2_CH=CHCH_2_CH=CHCH_2_ CH=CHCH_2_CH=CHCH_2_CH=CHCH_2_CH=CH(CH_2_)_2_COOH	22:6*n*−3
Linoleic acid	(9Z,12Z)-octadeca-9,12- dienoicacid	CH_3_(CH_2_)_4_CH=CHCH_2_CH=CH(CH_2_)_7_COOH	18:2*n*−6
Arachidonic acid	(5Z,8Z,11Z,14Z)-icosa- 5,8,11,14-tetraenoic acid	CH_3_(CH_2_)_4_CH=CHCH_2_CH=CHCH_2_CH=CH CH_2_CH=CH(CH_2_)_3_COOH	20:4*n*−6

IUPAC, International Union of Pure and Applied Chemistry.

## Data Availability

Not applicable.

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
