# Peer review of "The Immunometabolic Roles of Various Fatty Acids in Macrophages and Lymphocytes"

_ijms, 2021, doi:10.3390/ijms22168460_

Round 1

Reviewer 1 Report

Neto JCR et. al. describe the influences of metabolic processes and perturbations in macrophages and lymphocytes, in particular the role of fatty acids in phenotype and function.  The text is generally of a scholarly standard and the authors have wisely chosen to highlight a specific area of fatty acid metabolism. 

1>Rationale
The authors have ambitiously attempted to include in their discussion macrophages and lymphocytes (principally CD4+ and CD8+ T cells and B cells).  The rationale for choosing these cell lineages is not made clear.  Why not 'dendritic cells', granulocytes or innate lymphoid cells?  Additionally, the somewhat disparate nature of Mphi and lymphocytes does not allow the authors to discuss the area deeply compared to a review limited to one cell type.  This lack of depth could be offset by a comparative discussion of the two lineages, contrasting and comparing their roles and the effect of FA; e.g. at which stage of the inflammatory response does each metabolic process predominate?  Even within each section focussing on each cell type there is a lack of narrative and strong conclusion.  Each section (Mphi/Lymphocytes) should at least have a concluding paragraph.

2>Warburg effect
At line 379 the authors state that 'glycolysis is rapidly activated to induce the Warburg effect' and reference Boothby and Rickert 2017.  
(a) This citation is for a review, which is preferentially cited for topics outside the area of discussion or generally accepted findings, but this is a specific point and they should cite the original research.
(b) Boothby and Rickerty 2017 makes no reference to the Warburg effect.
(c) The "activation of glycolysis (rapidly or otherwise) to induce the Warburg effect" is (at best) poorly phrased.  A switch to glycolysis contributes to the Warburg effect.
The authors should rephrase this section and consider the representation of the Warburg effect where mentioned elsewhere.

3> Some formatting from the editing process appears to remain in the manuscript e.g. Ln 27 highlighted semi-colons, Ln 215 "...(from which tissues / organs?)..."

Minor points
4>Ln 116 "...macrophage metabolism, please, see Curi et al...."

5>Ln 166"...and promoted impairment on Na/K-ATPase,..."

6>Ln 177 "Palmitic acid may stimulate macrophages directly via TLR-4." Citation needed.

7>Ln 197 Montenegro-Burke, Sutton et al. 2016 used macrophages derived from isolated monocytes not PBMC.

8>Ln 206 "Since omega-3 FAs are anti-inflammatory they could have a role in control of macrophage differentiation towards more inflammatory phenotypes".  This sounds contradictory.

9>The paragraph from Ln 315 to Ln 318 requires expansion and editing.

10>The sentence beginning Ln 361 "As for CD4 Treg..." does not have any relevance to the sentences prior to or after it, and is in the section discussing CD8+ lymphocytes.

Author Response

Report 1

 Comments and Suggestions for Authors

Neto JCR et. al. describe the influences of metabolic processes and perturbations in macrophages and lymphocytes, in particular the role of fatty acids in phenotype and function. The text is generally of a scholarly standard and the authors have wisely chosen to highlight a specific area of fatty acid metabolism. 

Authors’comments: We appreciate the considerations and suggestions. Comments are made in response to each point raised.

1>Rationale
The authors have ambitiously attempted to include in their discussion macrophages and lymphocytes (principally CD4+ and CD8+ T cells and B cells).  The rationale for choosing these cell lineages is not made clear.  Why not 'dendritic cells', granulocytes or innate lymphoid cells?  Additionally, the somewhat disparate nature of Mphi and lymphocytes does not allow the authors to discuss the area deeply compared to a review limited to one cell type.  This lack of depth could be offset by a comparative discussion of the two lineages, contrasting and comparing their roles and the effect of FA; e.g. at which stage of the inflammatory response does each metabolic process predominate?  Even within each section focussing on each cell type there is a lack of narrative and strong conclusion.  Each section (Mphi/Lymphocytes) should at least have a concluding paragraph.

Authors’ comments: Our aim in this manuscript was to present a broad picture about fatty acid metabolism in immune cells. Unfortunately, studies regarding fatty acid metabolism in neutrophils or other granulocytes are scarce. Moreover, the fatty acid metabolism in dendritic cells is similar to what is found in macrophages. In innate lymphoid cells the studies about metabolism are interesting, although the effects on fatty acid metabolism are scarce too. Thus, we chose to focus on macrophages and lymphocytes since most is known about these cell types. The suggestions about concluding remarks are accepted and relevant comments are now added. 

2>Warburg effect
At line 379 the authors state that 'glycolysis is rapidly activated to induce the Warburg effect' and reference Boothby and Rickert 2017.  
(a) This citation is for a review, which is preferentially cited for topics outside the area of discussion or generally accepted findings, but this is a specific point and they should cite the original research.
(b) Boothby and Rickerty 2017 makes no reference to the Warburg effect.
(c) The "activation of glycolysis (rapidly or otherwise) to induce the Warburg effect" is (at best) poorly phrased.  A switch to glycolysis contributes to the Warburg effect.
The authors should rephrase this section and consider the representation of the Warburg effect where mentioned elsewhere.

Authors’ comments: Thank you for these comments: we agree that we have been a little imprecise here. This part was rephrased as: “In plasma cells, that are crucial for antibody production, glycolysis and oxidative glucose metabolism are activated in order to support increased mitochondrial capacity, associated with the biosynthesis of antibodies” and the reference replaced.

3> Some formatting from the editing process appears to remain in the manuscript e.g. Ln 27 highlighted semi-colons, Ln 215 "...(from which tissues / organs?)..."

Authors’ comments: We have corrected this.

Minor points
4>Ln 116 "...macrophage metabolism, please, see Curi et al...."

Authors’ comments: the word “please” was deleted

5>Ln 166"...and promoted impairment on Na/K-ATPase,..."

Authors’ comments: “Na/K-ATPase” was replaced by “Na+/K+-ATPase”

6>Ln 177 "Palmitic acid may stimulate macrophages directly via TLR-4." Citation needed.

Authors’ comments: a reference was added

7>Ln 197 Montenegro-Burke, Sutton et al. 2016 used macrophages derived from isolated monocytes not PBMC.

Authors’ comments: Thank you, we corrected this.

8>Ln 206 "Since omega-3 FAs are anti-inflammatory they could have a role in control of macrophage differentiation towards more inflammatory phenotypes".  This sounds contradictory.

Authors’ comments: Sorry if our wording was not clear. The phrase was rewritten “Since omega-3 FAs are anti-inflammatory they could also have a role in attenuation of macrophage differentiation towards more inflammatory phenotypes"

9>The paragraph from Ln 315 to Ln 318 requires expansion and editing.

Authors’ comments: We excluded the role of LAL in Treg subsets because this is unclear. We extend the role of ACC and de novo lipogenesis in TH17.

10>The sentence beginning Ln 361 "As for CD4 Treg..." does not have any relevance to the sentences prior to or after it, and is in the section discussing CD8+ lymphocytes.

Authors’ comments: We agreed. It was deleted.

Reviewer 2 Report

The manuscript by Dr. Silveira and his colleague summarized the current knowledge of Fatty acids in macrophages and lymphocytes. The article is well written and provides thorough reviews on this important topic. A few minor points should be considered:

1. There are many grammar errors in the manuscript. Please correct these errors in the revised manuscript.

2. Authors may include these references:

Calder PC. The effects of fatty acids on lymphocyte functions. Braz J Med Biol Res. 1993 Sep;26(9):901-17. PMID: 8298526.

Hubler MJ, Kennedy AJ. Role of lipids in the metabolism and activation of immune cells. J Nutr Biochem. 2016;34:1-7. doi:10.1016/j.jnutbio.2015.11.002

Author Response

 Report 2

Open Review

Comments and Suggestions for Authors

The manuscript by Dr. Silveira and his colleague summarized the current knowledge of Fatty acids in macrophages and lymphocytes. The article is well written and provides thorough reviews on this important topic. A few minor points should be considered:

Authors’ comments: Many thanks for the compliments. The suggestions were accepted.

  1. There are many grammar errors in the manuscript. Please correct these errors in the revised manuscript.

Authors’ comments: The grammar was reviewed.

  1. Authors may include these references:

Calder PC. The effects of fatty acids on lymphocyte functions. Braz J Med Biol Res. 1993 Sep;26(9):901-17. PMID: 8298526.

Hubler MJ, Kennedy AJ. Role of lipids in the metabolism and activation of immune cells. J NutrBiochem. 2016;34:1-7. doi:10.1016/j.jnutbio.2015.11.002

Authors’ comments: The references were included. Calder in line 446 and Hubler in line 110.

Reviewer 3 Report

The manuscript ijms-1292261, The Immunometabolic Roles of various Fatty Acids in Macrophages and Lymphocytes, presents an interesting and generally well done review of literature.

The references are not presented according to the mdpi style, both in the manuscript body and as the final list. The figures are also presented in the wrong way, at the end of the paper. There are some other editorial mistakes that the authors should have checked before the submission and should be corrected now. It seems that the paper was prepared initially for another journal and the change of style was not completed.

The introduction could be improved by presenting and commenting on important reviews works on fatty acids effects. See for example:

Analysis of the intricate effects of polyunsaturated fatty acids and polyphenols on inflammatory pathways in health and disease, Food Chem Toxicol. 2020 Sep; 143: 111558

Polyunsaturated Fatty Acids and Their Potential Therapeutic Role in Cardiovascular System Disorders—A Review, Nutrients. 2018 Oct; 10(10): 1561

An Update on Omega-3 Polyunsaturated Fatty Acids and Cardiovascular Health, Nutrients. 2021 Jan; 13(1): 204

Based on these works, or other similar ones, the authors should detail in the introduction on the type of FA and better explain the meaning of n-3, n-6 and so on. See rows 171-173. Explain what does omega-3 mean or omega-6. Explain what PUFA means. The authors introduce the term without explaining it (row 237). Detail what short-chain fatty acids means.

The review would considerably benefit if the authors would add some tables to better convey the information to the readers. For example, I think it would be useful to add a table with the major types of FA mentioned and their chemical class, like saturated or monounsaturated or polyunsaturated.

On row 232 and similar case: add the dose that produced the mentioned effect.

On row 228, check “Palmitate“. Chemically is not the same as palmitic acid and seems incomplete. Usually it is used for a salt (like sodium palmitate) or an ester (like glyceryl palmitate).

The authors should detail on important pathways that are mentioned. See row 323 PTEN, row 375 PI3K-AKT-mTORC1. Explain how these are connected.

The concluding remarks section is underdeveloped. See “This could facilitate the development of new therapeutic approaches based on immunometabolic actions of FAs”. This sentence adds little information. What type of diseases are you referring to? What types of FAs? Limitations of this approach?

Author Response

Report 3

Open Review

Comments and Suggestions for Authors

The manuscript ijms-1292261, The Immunometabolic Roles of various Fatty Acids in Macrophages and Lymphocytes, presents an interesting and generally well done review of literature.

The references are not presented according to the mdpi style, both in the manuscript body and as the final list. The figures are also presented in the wrong way, at the end of the paper. There are some other editorial mistakes that the authors should have checked before the submission and should be corrected now. It seems that the paper was prepared initially for another journal and the change of style was not completed.

Authors’ comments: We appreciate all the comments and we have addressed these. We have corrected references and figures format according to MDPI requirements. We apologize for the submission in incorrect format, but assure the reviewer that this paper was prepared exclusively to be published in this Special Issue of IJMS.

The introduction could be improved by presenting and commenting on important reviews works on fatty acids effects. See for example:

Analysis of the intricate effects of polyunsaturated fatty acids and polyphenols on inflammatory pathways in health and disease, Food ChemToxicol. 2020 Sep; 143: 111558

Polyunsaturated Fatty Acids and Their Potential Therapeutic Role in Cardiovascular System Disorders—A Review, Nutrients. 2018 Oct; 10(10): 1561

An Update on Omega-3 Polyunsaturated Fatty Acids and Cardiovascular Health, Nutrients. 2021 Jan; 13(1): 204

Authors’ comment: The suggested references were added in the introduction.

Based on these works, or other similar ones, the authors should detail in the introduction on the type of FA and better explain the meaning of n-3, n-6 and so on. See rows 171-173. Explain what does omega-3 mean or omega-6. Explain what PUFA means. The authors introduce the term without explaining it (row 237). Detail what short-chain fatty acids means.

Authors’ comment: A brief description about fatty acids was included in the introduction.

The review would considerably benefit if the authors would add some tables to better convey the information to the readers. For example, I think it would be useful to add a table with the major types of FA mentioned and their chemical class, like saturated or monounsaturated or polyunsaturated.

Authors’ comment: Thank you, that is a great idea which we have incorporated.

On row 232 and similar case: add the dose that produced the mentioned effect.

Authors’ comment: The doses used in the studies are now added.

On row 228, check “Palmitate“. Chemically is not the same as palmitic acid and seems incomplete. Usually it is used for a salt (like sodium palmitate) or an ester (like glyceryl palmitate).

Authors’ comment: Yes, we agree. This term was changed to palmitic acid

The authors should detail on important pathways that are mentioned. See row 323 PTEN, row 375 PI3K-AKT-mTORC1. Explain how these are connected.

Authors’ comment: We have tried to improve these aspects.

The concluding remarks section is underdeveloped. See “This could facilitate the development of new therapeutic approaches based on immunometabolic actions of FAs”. This sentence adds little information. What type of diseases are you referring to? What types of FAs? Limitations of this approach?

Authors’ comment: We added more information on the last part of the conclusion.

Round 2

Reviewer 1 Report

The authors' reply has addressed the queries raised in the first round review.  Although the authors have adopted the journal style and format of IMJS, they have not highlighted changes throughout the revised manuscript, which therefore necessitated a thorough review of the amended document in order to discern differences.

In the reviewer's opinion, further editing may be necessary in order to elevate the mansuscript to a publishable standard.

The issues are primarily 1) unnecessary text, 2) typographical errors and 3) lapses in scholarship.

An example of 1) is found at Ln 84 to 95 where the authors state "T and B lymphocytes are... undergo rearrangement of the T cell receptor (TCR)."  
This text is unnecessary and would be assumed knowledge for science undergraduate students.
The authors should carefully evaluate the text is relevant to the subject.

Examples of 2) include:
Ln 21 "fuel based" fuel-based
Ln 51 "...group[8] The names, structures..."
Ln 52 "table 1" should be Table 1.
Ln 141 Macrophages metabolism.  Rephrase or insert an apostrophe.
Ln 141 "Classicaly activated macrophages"  Classically-activated macrophages.  Ensure hypen usage is uniform.
The authors should carefully review the text to eliminate errors.

Examples of 3) include: 
Ln 130-132 where the authors assert "the overexpression of glucose transporter-1 (GLUT-1) can induce classical activation of monocytes to the M1 subtype" and cite Ref 29, Sundaram S et al (2014) which does not include reference to Glut1 or monocytes.  One of the authors on this paper (Freemerman AJ) is the first author on PMID: 24492615 which has more relevance to the cited sentence. 
Ref 33 (Ln 136) also appears to be erroneous - perhaps Ref 30 is intended?

In the first round review of this manuscript the authors cited a paper in reference to the Warburg effect, which had no relationship to the Warburg effect.  Correctly citing a published work is the foundation of scholarly works, and citations are analysed to determine impact factors for journals and the H-index for academics.  This is particularly the case for literature reviews which are essentially an analysis of published literature.  
Before resumitting the manuscript, the authors must consider that correct citations in a literature review are critical for acceptance.

Minor points
The abbreviation for short-chain fatty acids (SCFAs) is used twice; once when being defined and then two sentences later.  Perhaps this paragraph could be restructured to obviate the need for abbreviation?

Th1 is defined (Ln112) after the first use (89).

At line 67 and 68 the authors state that M1/M2 are the former names of polarised macrophages.  Please cite the reference denoting the correct nomenclature.

Author Response

The authors' reply has addressed the queries raised in the first round review.  Although the authors have adopted the journal style and format of IMJS, they have not highlighted changes throughout the revised manuscript, which therefore necessitated a thorough review of the amended document in order to discern differences.

In the reviewer's opinion, further editing may be necessary in order to elevate the mansuscript to a publishable standard.

The issues are primarily 1) unnecessary text, 2) typographical errors and 3) lapses in scholarship.

An example of 1) is found at Ln 84 to 95 where the authors state "T and B lymphocytes are... undergo rearrangement of the T cell receptor (TCR)."  
This text is unnecessary and would be assumed knowledge for science undergraduate students.
The authors should carefully evaluate the text is relevant to the subject.

Response: We have modified the text throughout the manuscript in accordance with this advice. However, some “basic” statements are necessary in order to link to the metabolic effects described. Also, the view of the reviewer would be supported if ALL science undergraduate students studied immunology – this may not be the case. Further, it is worth noting that another reviewer previously requested that we add information describing fatty acid naming and structure and insert a table with this information - this is also basic textbook information. Thus it is evident that different reviewers might have different views on the nature of basic information to include.

Examples of 2) include:
Ln 21 "fuel based" fuel-based
Ln 51 "...group[8] The names, structures..."
Ln 52 "table 1" should be Table 1.
Ln 141 Macrophages metabolism.  Rephrase or insert an apostrophe.
Ln 141 "Classicaly activated macrophages"  Classically-activated macrophages.  Ensure hypen usage is uniform.
The authors should carefully review the text to eliminate errors.

Response: We have corrected such errors.

Examples of 3) include: 
Ln 130-132 where the authors assert "the overexpression of glucose transporter-1 (GLUT-1) can induce classical activation of monocytes to the M1 subtype" and cite Ref 29, Sundaram S et al (2014) which does not include reference to Glut1 or monocytes.  One of the authors on this paper (Freemerman AJ) is the first author on PMID: 24492615 which has more relevance to the cited sentence. 
Ref 33 (Ln 136) also appears to be erroneous - perhaps Ref 30 is intended?

Response: We have exchanged the Sundaram reference for the Freemernan one. Reference 33 is correctly cited: it contains a significant discussion of fatty acid handling and metabolism in different macrophage phenotypes.

In the first round review of this manuscript the authors cited a paper in reference to the Warburg effect, which had no relationship to the Warburg effect.  Correctly citing a published work is the foundation of scholarly works, and citations are analysed to determine impact factors for journals and the H-index for academics.  This is particularly the case for literature reviews which are essentially an analysis of published literature.  
Before resumitting the manuscript, the authors must consider that correct citations in a literature review are critical for acceptance.

Response: We have checked that all references are appropriate for citation according to the statement requiring support. Reference 83 explicitly refers to and describes the Warburg effect.

Minor points
The abbreviation for short-chain fatty acids (SCFAs) is used twice; once when being defined and then two sentences later.  Perhaps this paragraph could be restructured to obviate the need for abbreviation?

Response: This is now done

Th1 is defined (Ln112) after the first use (89).

Response: Done

At line 67 and 68 the authors state that M1/M2 are the former names of polarised macrophages.  Please cite the reference denoting the correct nomenclature.

Response: Existing References 13 and 14 address this as does the reference previously numbered as 28 and now cited as reference 15.

Reviewer 3 Report

Even if the corrections performed by the authors are not highlighted, it seems that they responded to all the suggestions mentioned in the review and improved their paper considerably.

Author Response

Sorry by the inconvenience of submitting manuscript without the track changes. 

Thanks for suggestions.